# Genome-Wide Identification and Stress Responses of Cowpea Thaumatin-like Proteins: A Comprehensive Analysis

**DOI:** 10.3390/plants13223245

**Published:** 2024-11-19

**Authors:** Carolline de Jesús-Pires, José Ribamar Costa Ferreira-Neto, Roberta Lane de Oliveira-Silva, Jéssica Barboza da Silva, Manassés Daniel da Silva, Antônio Félix da Costa, Ana Maria Benko-Iseppon

**Affiliations:** 1Laboratory of Plant Genetics and Biotechnology, Center of Biosciences, Genetics Department, Federal University of Pernambuco, Av. Prof. Moraes Rego, 1235, Recife 50670-901, PE, Brazil; carollinejpires@gmail.com (C.d.J.-P.); lane.roberta@gmail.com (R.L.d.O.-S.); jessica.barboza@ufpe.br (J.B.d.S.); manasses.dsilva@ufpe.br (M.D.d.S.); 2Pernambuco Agronomic Institute, Av. Gen. San Martin, 1371-Bongi, Recife 50761-000, PE, Brazil; felix.antonio@ipa.br

**Keywords:** PR-5 gene family, osmotin-like, TLPs, *Vigna unguiculata*, zeamatin

## Abstract

Cowpea (*Vigna unguiculata* (L.) Walp.) is an important legume cultivated mainly in regions with limited water availability across the African and American continents. Its productivity is significantly affected by environmental stresses. Thaumatin-like proteins (TLPs), which belong to the PR-5 (pathogenesis-related 5) protein family, are known to be responsive to both biotic and abiotic stresses. However, their role remains controversial, with some TLPs associated with plant defense (particularly against fungal infections) and others associated with abiotic stresses response. In this study, we evaluated the structural diversity and gene expression of TLPs in cowpea (VuTLPs) under different stress conditions, including biotic [mechanical injury followed by inoculation with *Cowpea Aphid-borne Mosaic Virus* (CABMV) or *Cowpea Severe Mosaic Virus* (CPSMV)] and abiotic (root dehydration). Genomic anchoring of VuTLPs revealed 34 loci encoding these proteins. Neighbor- joining analysis clustered the VuTLPs into three distinct groups. We identified 15 segmental duplication and 6 tandem duplication gene pairs, with the majority of VuTLP genes found to be under purifying selection. Promoter analysis associated VuTLPs with bHLH, Dof-type, and MYB- related transcription factors, supporting their diverse roles. Diversity in VuTLP function was also observed in their expression profiles under the studied stress conditions. Gene expression data showed that most VuTLPs are recruited within the first minutes after biotic stress imposition. For the root dehydration assay, the most transcripts were up-regulated 150 min post-stress. Moreover, the gene expression data suggested that VuTLPs exhibit functional specialization depending on the stress condition, highlighting their diverse roles and biotechnological potential.

## 1. Introduction

In the course of the evolution, plants have developed a series of components to fight pathogens. Among several protein families involved in plant defense, we single out thaumatin-like proteins (TLPs), which are members of the pathogenesis-related 5 (PR-5) gene family. TLPs derive their name from their structural similarity to thaumatins, which are proteins isolated from the fruit of *Thaumatococcus daniellii* Benth. (Maranthaceae family) [1]. TLPs have the thaumatin conserved domain (Pfam: PF00314) that covers almost 95% of the entire mature peptide [2]. Regarding their potential biological functions, the referred protein group have shown antifungal activity [3,4], acting on fungal membrane permeabilization [5]. There is evidence that some TLP representatives act on invading fungi through their binding activities or hydrolysis of β-1,3-glucans [6,7] or, still, through inhibition of enzymes such as xylanases [8].

Hormones associated with plant immunity can also induce TLP-coding genes. For example, in tobacco (*Nicotiana tabacum*) seedlings, the induction of TLP genes occurred in response to the application of ethylene (ET), methyl jasmonate (methylJA), and salicylic acid (SA) [9]. A similar observation was also reported for transgenic rice (*Oryza sativa* L. cv. Sasanishiki), where the rice blast fungus elicitor, besides the hormones SA and methylJA, up-regulated the β-glucuronidase (GUS) reporter gene driven by the Rtlp1 (rice thaumatin-like protein 1) promoter (pRtlp1GUS) [10].

TLPs expression are also associated with abiotic stresses as drought [11,12,13], wounding [14], and freezing [15,16]. There are, additionally, reports that TLPs were up-regulated after infection by viruses or bacteria [17]. However, in these situations, their action mechanism remains indeterminate. Regarding viruses, Kim et al. (2005) [18] observed that the tobacco NtTLP1 interacted with *Cucumber mosaic virus* proteins under both in vitro and in planta conditions. The NtTLP1 protein engaged in specific interaction with CMV proteins involved in viral movement, particularly the movement protein and the coat protein, during yeast-based experiments. Furthermore, when the authors employed the RT-qPCR technique to analyze gene expression, they observed an induction of NtTLP1 transcripts in response to CMV infection.

TLPs have garnered increasing attention for their biotechnological potential. Several studies have demonstrated that transgenic organisms overexpressing these proteins exhibit enhanced performance under various biotic and abiotic stress conditions [19,20,21]. The mentioned protein group exhibits a high degree of multifunctionality, making it an attractive target for gene transformation/editing studies and for analyzing its physiological impact in economically important organisms.

Due to their significance, TLPs have been analyzed in a large number of species, such as *Arabidopsis thaliana*, rice, *Populus* spp., maize (*Zea mays*), *Physcomitrella patens*, *Chlamydomonas* spp., and wheat (*Triticum aestivum*) [12,22,23,24], among others. However, information regarding this protein class in legumes remains quite limited. The vast majority of the studies for the mentioned clade do not explicitly focus on TLPs’ global analysis, but only quote some TLP participation in the global transcriptional response or focus on specific TLP members. The only report found in the literature addressing the structural and functional genomics of legume TLPs was performed by Petre et al. [2]. These authors identified 42 validated TLP-loci in the genome *Populus trichocarpa* “Nisqually-1”. In addition, it was observed that most TLPs were responsive to abiotic stresses (high ozone content, UV-B rays, drought, and high copper content) besides being also responsive to biotic stress (infection by the fungus *Melampsora laricis-populina*).

Cowpea is an important legume crop. Its social role is crucial: such crop is source of proteins and minerals for millions of people in sub-Saharan Africa and other developing regions [25]. Cowpea demonstrates exceptional phenotypic plasticity and strong physiological performance under a wide array of biotic and abiotic stresses. Given its importance and adaptive capacity, considerable efforts in omics research have been directed toward this species. Recently, its reference genome was released [26], providing a vital resource for elucidating the crop’s physiological advantages under stress conditions, identifying genes with biotechnological potential, and serving as a key platform for comparative genomics within legume clade. In Brazil—a country that contributes 12% of global cowpea production—the Cowpea Genomics Consortium (CpGC) was established. The CpGC manages sequenced genomes from multiple accessions and cultivars of the referred species. In addition, it contains comprehensive transcriptomes (RNA-Seq) from cultivars that demonstrate tolerance or resistance to stresses. These cultivars have been subjected to root dehydration or combined stressors, such as mechanical injury followed by viral inoculation with either *Cowpea Aphid-borne Mosaic Virus* or *Cowpea Severe Mosaic Virus*. These data have contributed to a better understanding of the phenotypic robustness of this important crop.

As climate change and the demand for sustainable agriculture intensify, understanding TLPs in cowpea can provide valuable insights into its defense and tolerance strategies. This knowledge can support the development of genetically modified cultivars with enhanced performance under unfavorable conditions, ultimately improving yield and food security. By studying TLP expression patterns in cowpea under various stress types, we can analyze their regulatory mechanisms and potential roles in stress adaptation. Investigating TLPs in the referred species not only contributes to our understanding of plant defense mechanisms but also holds promise for biotechnological advancements in this strategic legume crop.

Thus, given the importance, multifunctionality, the responsiveness of plant TLPs to unfavorable conditions and the availability of CpGC data, the present work brings structural characterization and gene expression data of cowpea TLPs (VuTLPs) expressed under different stress types (mechanical injury followed by CABMV or CPSMV inoculation, and root dehydration). The present work uncovered relevant information to the molecular physiology of cowpea under unfavorable conditions and provide data about the performance of TLPs in a legume species, an issue still little addressed.

## 2. Results

### 2.1. In Silico Identification and Characterization of VuTLPs

The tBLASTn search allowed for the identification of 126 putative VuTLPs (Appendix A) expressed under biotic and abiotic stresses. From these, 33 presented no ORF associated with TLP, while 32 presented with a complete TLP conserved domain (CTD), and 61 presented with an incomplete CTD. Among the candidate VuTLPs with complete CTD, 30 were expressed in cowpea RNA-Seq libraries generated under radicular dehydration, while 28 and 27 VuTLPs were expressed under mechanical injury followed by CABMV or CPSMV inoculation, respectively. Considering the total amount (abiotic + biotic stresses), 30 non-redundant VuTLPs were retrieved (Appendix A). Information on the VuTLP identifiers (loci, RNA-Seq transcripts, and proteins) and their corresponding counterparts in the cowpea genome (Vigna_unguiculata_469_v1.2), available in the Phytozome database, is provided in Appendix A.

All 30 predicted VuTLPs presented CTD (Appendix A). Five highly conserved amino acids (REDDD motif) in an acidic cleft were detected in most VuTLPs. However, the alignment of the candidates with other TLPs from *A. thaliana*, tobacco, maize (*Zea mays*), and chickpea (*Cicer arietinum*) revealed that some VuTLPs did not contain all expected amino acids in REDDD motif. In this group, some members exhibited the following substitutions (Appendix A): glutamic acid (E) was replaced by glutamine (Q) in the VuTLP32.1 transcript; in the VuTLP12.2 and VuTLP12.1 transcripts, glutamic acid (E) was replaced by aspartic acid (D); and the first aspartic acid (D) was replaced by glycine (G) in the VuTLP32.1 transcript. Other modifications in the mentioned motif were also observed. For instance, in the VuTLP12.2 and VuTLP12.1 transcripts (Appendix A), the second aspartic acid (D) was replaced by asparagine (N).

Regarding the 16-cysteine conservation, just three cowpea sequences (VuTLP26.1, VuTLP13.2 and VuTLP13.1; Appendix A) lacked the first cysteine, whereas the sixth cysteine from the transcript VuTLP11.1 was replaced by a serine (S). All the other VuTLPs presented the 16 mentioned residues. Additionally, the VuTLPs exhibited a molecular weight ranging between 18.97 and 34.93 kDa and an isoelectric point varying from 4.23 to 9.05 (Appendix A).

The SignalP prediction, in turn, revealed that 20 VuTLPs present hydrophobic signal peptide sequences (Appendix A).

### 2.2. Mapping of VuTLPs in V. unguiculata Genome and Analysis of Gene Duplication Mechanisms

The applied approach revealed the distribution of VuTLPs in 34 loci located along all the cowpea chromosomes, with exception of chromosome 10 (Figure 1).

A clustering of VuTLPs loci was observed, being more pronounced on chromosomes two and eight (Figure 1; Appendix A). For example, the long arm of cowpea chromosome “2” presented six and two clustered TLP-coding genes (Figure 1). Such a distribution is indicative of evolutionary processes through tandem duplication. Besides the observed clustering in most chromosomes, the position of VuTLP-coding genes was mainly subterminal or intercalary, while a pericentric position of VuTLPs is observed only in chromosomes “6” and “7” (Figure 1).

Gene duplications, including segmental and tandem duplication, have been considered as one of the main forces in the evolution and expansion of a gene family. The analysis of the VuTLP gene family expansion revealed that the 34 TLP-coding genes have expanded through different gene duplication processes (Appendix A). In this study, we found four types of gene duplications mechanisms: dispersed, proximal, tandem, and WGD or segmental duplication (Appendix A). Our analysis provides evidence that whole-genome duplication (WGD) or segmental duplication has been the primary mechanism driving the expansion of VuTLPs, resulting in a total of 15 gene pairs (Appendix A). Moreover, VuTLP genes were observed to cluster within six tandem duplication regions across 4 of the 11 chromosomes (Appendix A). Finally, we identified five dispersed duplication events and one proximal duplication event contributing to the expansion of these genes (Appendix A).

### 2.3. Ratio of Synonymous and Non-Synonymous Substitutions for Tandem Duplicated Genes

The forces driving natural selection can be inferred from the types of nucleotide substitutions in the coding sequences of genes. A key parameter for understanding the evolution of genes under selection is the ratio of nonsynonymous substitution rates (Ka—those that cause amino acid changes) to synonymous substitution rates (Ks—those that do not cause amino acid changes). Ka/Ks values ranged from 0.00 to 1.36 for tandem duplications (Appendix A) and from 0.00 to 0.31 for segmental duplications (Appendix A). Sixteen VuTLP gene pairs presented a Ka/Ks ratio of less than 1 (Appendix A), suggesting that these genes have been subjected to purifying selection. In a different context, it was observed that five VuTLP gene pairs exhibited Ka/Ks values greater than 1 (Appendix A), indicating that positive selection played a significant role in the evolution of these pairs.

### 2.4. NJ Analysis and Exon–Intron Organization

A neighbor-joining tree was constructed using the protein sequences of all the 34 VuTLP genes. The cowpea VuTLP family was divided into three groups (Appendix A). Group III, colored in purple (58.82%), contained the most members; followed by Group II, colored in pink (27.27%); and the least represented group was the I, colored in green (5.88%), with only two VuTLPs (VuTLP16 and VuTLP32) (Appendix A). To understand the formation of these gene groups, several characteristics were analyzed, including the presence of a signal peptide, molecular weight, isoelectric point, REDDD residue presence, and localization (Table 1). Group I comprised genes with molecular weights ranging from 24.16 to 25.18 kDa and isoelectric points from 6.80 to 7.32. Notably, in the gene VuTLP32, alterations were observed in the REDDD residues, where E was replaced by Q, and the first D was replaced by G. Group II included genes with molecular weights ranging from 21.48 to 29.20 kDa and isoelectric points from 4.04 to 8.76. In the gene VuTLP12, a change in the REDDD residues was noted, with the second D replaced by N. Group III consisted of genes with molecular weights from 13.75 to 108.18 kDa and isoelectric points from 4.00 to 8.19. In the gene VuTLP6, the REDDD residue showed an alteration where R was replaced by Q. Additionally, in the genes VuTLP7, VuTLP5, and VuTLP18, the first D residue was substituted by Y, and in VuTLP18, the second D was replaced by Q. Moreover, within this group, two genes (VuTLP5 and VuTLP18) were predicted to localize to the plasma membrane.

Additionally, most of the 34 VuTLP genes analyzed in the gene structure analysis contained three exons; while VuTLP7, VuTLP31, VuTLP20, VuTLP21, VuTLP29, VuTLP1, VuTLP17, VuTLP15, VuTLP24, VuTLP25, and VuTLP16 had two exons; and VuTLP4, VuTLP19, VuTLP9, VuTLP11, VuTLP8, VuTLP10, and VuTLP32 had one exon. Four genes (VuTLP11, VuTLP8, VuTLP10 and VuTLP32) had no introns (Appendix A).

### 2.5. Candidate Cis-Regulatory Element Analysis

The 34 *VuTLP* genes had their predicted promoter regions (1 kb) analyzed regarding the presence and identity of candidate cis-regulatory elements (CCREs). Four of the identified CCREs were within the stipulated cut-off (e-value < 10−2, Figure 2; Appendix A), being considered as bona fide CCREs. The six remaining motifs (Figure 2) were associated with five different TFs [Dof-type (dark green and pink boxes), C2H2 (light green), G2-like (orange), and CPP (light blue), and one without specific annotation] but did not achieve the specified validation parameter (e-value < 10^−2^), thus being excluded from this study.

The identified bona fide CCREs were associated with three TF families (Figure 2): bHLH (two different motifs; light green and blue boxes, Appendix A), MYB-related (light blue motifs, Figure 2; Appendix A) and Dof-type (red motifs, Figure 2; Appendix A). Bona fide CCREs associated with a Dof-type TF (JASPAR ID MA1267.1; Figure 2; Appendix A) were the most abundant (observed in 42 sites) in the analyzed promoter regions, followed by bona fide CCREs associated with MYB-related (JASPAR ID MA1182.1; Figure 2; Appendix A) and bHLH (JASPAR ID MA0587.1; Figure 2; Appendix A) TFs, each anchored in 11 sites. Finally, another bona fide CCRE associated with a bHLH TF (JASPAR ID 1361.1; Figure 2; Appendix A) anchored in five sites.

### 2.6. TLP Content and Expression in Cowpea Transcriptomes Under Different Stress Types

Considering all available RNA-Seq libraries for biotic and abiotic stresses (Appendix A), 30 VuTLPs were identified. From this point onward, we will systematically divide the assays to provide a more detailed examination of VuTLP regulation.

#### 2.6.1. Mechanical Injury (MI) and Virus Inoculation Assays

Twenty-eight VuTLP-coding transcripts were expressed in the MI_CABMV RNA-Seq libraries (Appendix A). Of these, six VuTLPs (VuTLP10.1, VuTLP27.2, VuTLP12.2, VuTLP16.1, VuTLP26.1, and VuTLP11.1) were up-regulated exclusively at 60 min (Appendix A), suggesting that VuTLPs are preferably recruited during the first hours after the MI_CABMV treatments. In addition, only one VuTLP-encoding transcript (VuTLP9.1) showed up-regulation at 16 h. Additionally, the VuTLP3.1 transcript was up-regulated at 60 min and down-regulated at 16 h treatment, whereas the others presented constitutive expression or were not expressed in the comparisons.

Regarding the MI_CPSMV assay, twenty-seven VuTLP-coding transcripts were expressed (Appendix A). Of these, five were up-regulated exclusively at 60 min (VuTLP28.1, VuTLP33.1, VuTLP3.1, VuTLP25.1, and VuTLP16.1) and three at 16 h (VuTLP13.2, VuTLP22.2, and VuTLP9.1) (Appendix A). In addition, VuTLP11.1 transcript was up-regulated in both treatments (60 min and 16 h; Appendix A). These results also point out that VuTLPs were recruited in the first hour after MI_CPSMV treatment.

When comparing the transcriptional orchestration in response to the two MI_viral inoculation assays, it was observed that four VuTLPs (VuTLP3.1, VuTLP16.1, VuTLP9.1, and VuTLP11.1) were up-regulated in response to both assays (Appendix A). In turn, four (VuTLP10.1, VuTLP27.2, VuTLP12.2, and VuTLP26.1) and five (VuTLP28.1, VuTLP33.1, VuTLP25.1, VuTLP13.2, and VuTLP22.2) VuTLP-coding transcripts were up-regulated exclusively in response to MI_CABMV or MI_CPSMV, respectively (Appendix A).

#### 2.6.2. Root Dehydration Assay

Thirty potential VuTLP-coding transcripts were expressed in cowpea under root dehydration (Appendix A). Of this quantity, VuTLP23.1 and VuTLP13.1 were up- and down-regulated at 25 min, respectively. The VuTLP30.1 transcript was up- regulated at 25 min and down-regulated at 150 min of treatment. In addition, the VuTLP22.2 transcript was up-regulated in both treatment times (25 min and 150 min). Eight (VuTLP22.1, VuTLP31.1, VuTLP27.2, VuTLP27.1, VuTLP2.1, VuTLP3.1, VuTLP8.1, and VuTLP11.1) and three (VuTLP32.1, VuTLP10.1, and VuTLP14.2) VuTLP-coding transcripts were up- and down-regulated exclusively at 150 min of treatment, respectively. These results indicate that some VuTLPs are recruited in the first minutes after stress imposition, even though a higher number of transcripts were detected at 150 min treatment (Appendix A).

#### 2.6.3. MI_CABMV vs. MI_CPSMV vs. Root Dehydration Assays

A comparison between the transcriptional orchestration in response to both MI_viruses (in leaves) and root dehydration (roots) assays revealed interesting insights. There were 20 VuTLPs expressed jointly in the assays (MI_CABMV and MI_CPSMV and root dehydration; Appendix A).

Qualitative analyses showed a heterogeneous orchestration of VuTLPs. Only two VuTLP candidates (VuTLP3.1 and VuTLP11.1) were up-regulated in the three analyzed conditions (Appendix A). Exclusive responses were also observed under different treatments. For example, three (VuTLP10.1, VuTLP12.2, and VuTLP26.1), four (VuTLP28.1, VuTLP33.1, VuTLP13.2, and VuTLP25.1), and seven (VuTLP22.1, VuTLP31.1, VuTLP30.1, VuTLP27.1, VuTLP2.1, VuTLP23.1, and VuTLP8.1) VuTLP-coding transcripts were up-regulated exclusively under MI_CABMV and MI_CPSMV inoculation and root dehydration, respectively. Thus, there is no overall conservation in VuTLP transcriptional orchestration towards different analyzed assays.

### 2.7. RNA-Seq Data Validation by qPCR

#### 2.7.1. MI_CABMV and MI_CPSMV Assays

The selection of a sample of target transcripts for qPCR data validation was based on their regulation in the RNA-Seq libraries (Appendix A) for MI_CABMV and MI_CPSMV experiments.

Considering the functional primer pairs, all demonstrated efficiency within acceptable standards (Appendix A). Relative expression analysis showed that VuTLP3.1, VuTLP11.1, VuTLP27.2, and VuTLP26.1 were up-regulated at 60 min (Appendix A) while VuTLP9.1 was up-regulated at 16 h (Appendix A), confirming RNA-Seq data for MI_CABMV assay. However, qPCR results have shown that VuTLP3.1 was not modulated at 16 h (not significant—ns), differing from RNA-Seq results, where it was down-regulated (Appendix A).

For the MI_CPSMV treatments, three differentially expressed target transcripts (VuTLP3.1, VuTLP22.2, and VuTLP9.1) were evaluated. All primer pairs analyzed in this step demonstrated efficiency within acceptable standards (Appendix A). VuTLP3.1 was up-regulated at 60 min and VuTLP22.2 and VuTLP9.1 were up-regulated at 16 h (Appendix A). All these results confirmed RNA-Seq expression data.

#### 2.7.2. Root Dehydration

For the mentioned assay, six differentially expressed target transcripts (VuTLP22.2, VuTLP3.1, VuTLP10.1, VuTLP32.1, VuTLP14.2, and VuTLP2.1) were analyzed by qPCR, all with acceptable efficiency values (93.00 to 107.70%; Appendix A). Relative expression analysis showed that VuTLP22.2, VuTLP3.1, and VuTLP2.1 were up-regulated at 150 min and VuTLP10.1 and VuTLP32.1 transcripts were down-regulated at 150 min (Appendix A). All these results confirmed RNA-Seq expression data.

## 3. Discussion

Using TLP sequences from other species as seed sequences, VuTLP candidates were searched in the cowpea genome and in cowpea RNA-Seq libraries generated under root dehydration and mechanical injury followed by CABMV or CPSMV inoculation. Such conditions were chosen to exemplify the participation of VuTLP in response to both biotic and abiotic stresses, since TLP participation in response to abiotic stress is still incipient when compared to its biotic counterpart. Additionally, RNA-Seq reads have up to 400 bp, depending on the sequencing technology used [27], allowing for direct VuTLP transcripts structural characterization.

Most of the identified VuTLPs presented the REDDD motif in an acidic cleft. This configuration is responsible for the reported antifungal activity of TLPs in plants [23,28]. However, the alignment of the referred VuTLP candidates with other TLPs from *A. thaliana*, tobacco, maize, and chickpea revealed that some VuTLPs did not contain all expected amino acids in the REDDD motif. It is still not clear whether these small differences have a significant impact on the substrate selectivity or protein function [2]. Petre et al. [2] found that the acidic cleft is the most conserved region among eukaryotic TLPs. For example, in sTLPs (small TLPs), the most of the REDDD amino acids are conserved, but wheat sTLP sequences with a resolved structure have revealed no acidic clefts nor any particular conserved region, which may be linked to the xylanase inhibitor function reported for such proteins [2,8].

Regarding the 16-cysteine conservation, with the exception of some VuTLPs, all other presented the 16 residues that form eight disulfide bonds necessary for correct folding and to ensure a high level of thermostability and pH constancy [29]. Most TLPs present molecular weights ranging from 21 to 26 kDa [30] and isoelectric point ranging between 3.4 and 12.0 [31], having this last one influence on the electrostatic potential at the molecular surface [32]. Thus, our results show that most of the VuTLPs presented values within the expected standard for both characteristics, except for some VuTLPs that presented specificities regarding their molecular weight.

The SignalP prediction revealed that most VuTLPs present hydrophobic signal peptide sequences, indicating that they are predominantly secreted, a result confirmed by TargetP prediction. TLPs are located mainly in two subcellular compartments, apoplastic space or vacuole, because of an N-terminal signal peptide that directs the mature protein to a secretory pathway and, in some instances, a C-terminal polypeptide that directs them to a vacuolar compartment [33]. Previous studies have used the GFP-gene approach, associated with a transient expression system, to study the localization of these proteins. Most of them point to a location in extracellular spaces [4,34,35], corroborating with our in silico prediction.

To infer on the distribution, relative position, and abundance of VuTLPs, we anchored candidate transcripts against the cowpea genome [26]. The number of VuTLPs is similar to those found in other leguminous plant genomes. Specifically, *Medicago truncatula* has 49, *Trifolium pratense* has 23, and *Lupinus albus* has 29 TLPs [15].

The results of gene duplication analysis suggested the existence of VuTLP duplication events in cowpea genome. VuTLP genes were found to cluster into six tandem duplicated regions on 4 of 11 chromosomes, implying that they may originate from the recent gene duplication events. Moreover, segmental duplications were common in the VuTLP family. Fifteen segmental duplication gene pairs were identified in cowpea genome—compared to three, two, seven, and four in *A. thaliana*, rice, poplar, and maize [23]—suggesting that a majority of the VuTLP genes were generated by the mentioned mechanism. Cao et al. [23] suggested that tandem and segmental duplications are the main factors leading to TLP family enlargement and diversity in plants. Yang et al. [4] identified six segmentally duplicated TLP gene pairs in the *Panax notoginseng* genome. TLP gene duplication analysis revealed thirteen gene duplication events in the watermelon genome, including one tandem and twelve segmental duplications [36]. Liu et al. [37] identified twelve segmentally duplicated and seven tandemly duplicated gene pairs that have experienced purifying selection in the melon genome.

Ka/Ks analysis is widely used to estimate how natural pressure and evolutionary forces affect duplicated genes and the corresponding proteins [38]. In cowpea, the majority of TLP genes were shown to be driven by purifying selection and a few sites of TLP sequences were shown to be driven by positive selection. Similar results were found in the study on the evolution of TLPs in A. thaliana, rice, and maize, which revealed that many TLP sequences were under purifying selection pressure and only on a few sites of TLP sequences may have experienced positive selection during the evolutionary process [23]. A study on poplar TLP genes suggested that four TLP kinases and ten TLPs had undergone some positive selection, indicating that plant TLPs may have experienced a diversified natural selection process [29]. Some of the exposed amino acids of the TLP structure are under positive selection, the amino acids that form the acidic cleft are always under purifying selection, indicating that conservation of the acidic cleft is crucial and could be important for the antifungal activity of TLPs [39].

In the current study, 34 VuTLP genes formed three groups through the NJ tree. In all the obtained groups, a clear distinction emerged regarding the VuTLP genes when analyzing the REDDD motif associated with duplication events. Group III, which contained the highest number of genes, displayed a consistent pattern of residue substitution in the REDDD motif, where the first D was replaced by Y. In contrast, genes in the other groups exhibited different substitution patterns. When comparing the conservation and substitution patterns of the REDDD residues with the duplication events, it was observed that residues maintaining conservation were expanded through segmental duplication and were under purifying selection. This indicates that these residues, and consequently the antifungal activity associated with this motif, were preserved. Conversely, residues that underwent alterations or substitutions expanded through tandem duplication and were subject to positive selection, suggesting a diversification of these residues that could lead to the emergence of new functions. Analyzing eukaryotic TLPs, Liu, Sturrock, and Ekramoddoullah [29] identified the formation of nine highly diversified groups, referring to the TLP superfamily. A possible explanation for the formation of these groups can be attributed to the structural diversity presented by TLPs. According to Petre et al. [2], such diversity may influence the biological and biochemical functions, and differences in the topology around the cleft could determine the specificity of TLPs to their target ligands [40]. This may be true also considering the VuTLPs since their diversity and position in different positions of the NJ tree is possibly due to the presence of orthologs and paralogs, with structural and functional specificities. Zhao et al. [11] identified four distinct groupings within the TLP gene family and highlighted the presence of distinct subfamilies within each group indicating the close evolutionary associations between certain TLP genes from faba bean and other legume species.

Over time, plants acquired new weapons to promote resistance [41,42] while pathogens also continued to evolve, bringing pressure on the structure and diversity of some associated gene families, increasing the size of some protein families [43], including PR-genes. Such an increase represents major significance for functional diversity via the sub- or neo-functionalization of paralogs [44]. In this scenario, the natural selection of genes with new functions under environmental pressure probably also played a significant role in the evolution of TLPs [29], as may also be suggested for cowpea.

In addition, regarding the gene structure of VuTLPs, genes with three, two, and one exon were verified, beside four genes that did not present introns. The organization of introns and exons of TLPs have been reported to be in the range of one to ten exons [2,23,29]. Out of the ten TLP genes identified in the faba bean transcriptome, two genes exhibited three exons, five genes consisted of only one exon, and three genes displayed two exons [11]. Yan et al. [45] found that most of the grape TLP genes had less than four exons and four genes also had no introns.

The 34 VuTLP-coding genes had their predicted promoters (1 kb) analyzed in regard to the presence and identity of CCREs. Discovered bona fide CCREs can be used to identify potential physiological or adaptive processes in which TLPs participate, associating transcription factors and their intrinsic biological processes. This action adds value to the molecular dynamics of cowpea TLPs, especially in association with the considered transcriptomic data.

Concerning the analyzed promoter regions, CCREs associated with bHLH, Dof-type, and MYB-related TFs attended the stipulated statistical parameters. Considering the literature data, these TFs present an intimate association with plant stress response and other physiological processes. bHLH members can act as transcriptional activator or repressor and play essential roles in metabolic and developmental processes [46]. Some bHLH members have also been reported to be modulated under cold, drought, and salt stress [47,48]. Dof-type transcription factors, in turn, participate widely in plant development and abiotic stress response [49,50]. In tobacco, the *Sar8.2b* gene can be activated by the Dof-type TF, which is related to systemic acquired resistance [51]. Finally, there are reports on MYB-related TFs acting as retrotransposon regulators and controlling defense-related genes, besides being induced by wounding and other elicitors, in tobacco plants [52]. Such in silico data on promoters associated with TLPs in RNA-Seq libraries under different stresses (biotic and abiotic) indicate a plurality of roles for TLPs, also reinforcing their biotechnological potential.

The comparison among the transcriptional orchestration in response to mechanical injury followed by viral inoculations (in leaf tissue) and root dehydration assays revealed VuTLPs expressed in the three assays. Additionally, there is no overall conservation in VuTLP transcriptional regulation towards different viral isolates and root dehydration (only two VuTLPs were up-regulated in all the studied stresses), suggesting that, primarily, specific VuTLPs are associated with the specific responses to each virus type and root dehydration.

Signalization response in plants submitted to abiotic and biotic stressors can induce separate and overlapping sets of genes, leading to the expression of distinct as well as common components [53,54]. These separate pathways show nodal points where they converge and crosstalk to optimize the various defense responses [55], resulting in shared stress mitigation strategy by combined morphophysiological processes and molecular responses [56]. The identification of crosstalk between biotic and abiotic signaling pathways (as the case of the two up-regulated VuTLPs in all the studied conditions) has been crucial for envisaging and strengthening our understanding of the regulation of plant response against combined stresses.

Several studies have shown that the transcriptional up-regulation of TLPs in plants has a positive effect on stress resistance and tolerance mechanisms. Singh et al. [57] analyzed an *Arachis diogoi* (AdTLP) TLP up-regulated under fungal infection (*Phaeoisariopsis personata*). The cDNA encoding AdTLP was cloned using RACE-PCR and used to transform tobacco plants. Overexpression of AdTLP resulted in increased resistance to pathogenic fungi and tolerance to abiotic stresses (high salinity and oxidative stress). In addition, transgenic plants also exhibited a higher level of transcription of genes *PR1a*, *PI-I*, and *PI-II* compared to *WT*. Such genes are associated with plant pathogen defense mechanisms. Chowdhury, Basu, and Kundu [19] showed that transgenic lines of sesame overexpressing an osmotin-like protein (SindOLP, a thaumatin-like protein) presented tolerance against abiotic stresses (drought and high salinity) and resistance against the fungus *Macrophomina phaseolina*. Overexpression of SindOLP resulted in the up-regulation of three genes [*superoxide dismutase* (*SiSOD*), *cysteine protease inhibitor* (*SiCysPI*), and *glutathione-S-transferase* (*SiGST*)] that encode enzymes for the elimination of reactive oxygen species (ROS), indicating that SindOLP participates in ROS regulation, which is common to both stresses addressed. Sun et al. [58], in turn, investigated transgenic Nanlin895 poplars expressing an overexpressed PeTLP gene [cloned from poplar cultivar (*Populus deltoides* × *P. euramericana* “Nanlin895”)]. While PeTLP alone did not directly inhibit pathogens, plants with higher PeTLP levels demonstrated increased resistance to spot disease. In vitro experiments revealed that leaf extracts from these transgenic plants suppressed fungal growth. These findings suggested that PeTLP acts as an inducer in transgenic Nanlin895 poplars, stimulating the production of other antifungal proteins within the plant. Finally, Cui et al. [21] explored the use of transgenic TaTLP1 wheat lines to bolster resistance against fungal pathogens *Bipolaris sorokiniana* (BS) and *Puccinia triticina* (PT). Transgenic lines overexpressing the TaTLP gene exhibited no significant differences in tiller number or 1000-kernel weight compared to the wild-type Jinan Wheat No. 1 (JW1). However, they demonstrated enhanced resistance to both studied pathogens. This increased resistance was likely attributed to the activation of peroxidase and β-1,3-glucanase enzymes following BS infection and the induction of reactive oxygen species-related genes after PT infection in the transgenic lines. These results suggested that stable TaTLP1 expression can effectively improve resistance to these fungal pathogens.

Some published papers have demonstrated the up-regulation of TLPs following virus inoculation, similar to what was observed in the present study. Madroñero et al. [59] analyzed changes in the papaya transcriptome in response to the inoculation with PMeV complex (*Papaya Meleira Virus*, PMeV, and *Papaya Meleira Virus 2*, PMeV2), which causes papaya sticky disease (PSD). To evaluate the effects of SA signaling on PMeV complex load, *Carica papaya* seedlings were exposed to this hormone and inoculated with both viruses. The SA-treated plants showed increased *PR1*, *PR5* (thaumatins-like proteins), and *CHIA* levels, confirming SA signaling activation. The same plants presented reduced PMeV complex load, suggesting that SA signaling and PR5 proteins play a role in *C. papaya* tolerance to PSD before flowering.

The RNA-Seq and qPCR results showed—despite being traditionally responsive to biotic stresses—TLPs were recruited under abiotic stress, as also seen in other works [2,11,12,13,14]. In the present study, a higher number of VuTLPs were up-regulated at 150 min after root dehydration. Zhao et al. [11] identified and characterized 10 TLP genes from the faba bean transcriptome. Two genes (VfTLP4-3 and VfTLP5) displayed high expression levels in the drought-tolerant cultivar when exposed to drought stress. In turn, Misra et al. [60] identified and characterized a new basil (*Ocimum basilicum*) TLP (ObTLP1) from ESTs recovered after MeJA treatment. The ectopic expression of ObTLP1 in *A. thaliana* provided fungal resistance (*Scleretonia sclerotiorum* and *B. cinerea*) and tolerance to drought and high salinity stresses.

Despite the results of the present work and previous studies, the mechanism of action of VuTLP proteins against abiotic stress remain largely unknown, although investigations have confirmed the participation of several TLP genes in cold temperature, salt, and drought stress responses [11,12,13,57,60]. Feng, Wei, and Li [61] investigated protein–protein interactions for AhTLP1 from *Arachis hypogaea* and GmOLPa/GmOLPb from soybean, aiming to establish more detailed potential functional pathways for these proteins. They identified two distinct clusters within the AhTLP1 interaction network. Cluster 1 included 10 proteins involved in glycerolipid and glycerophospholipid metabolism. Notably, protein phosphatase 2C, a key component of the ABA signaling pathway [62], was found to interact with AhTLP1. This pathway plays a crucial role in abiotic stress responses. Cluster 2 comprised 12 proteins associated with the endocytosis pathway. Considering protein–protein interaction analyses of GmOLPa and GmOLPb on the STRING database, it was revealed that GmOLPa is involved in sucrose biosynthesis, peroxisome organization, and carbohydrate metabolism. GmOLPb, on the other hand, is associated with sucrose biosynthesis and defense responses to other organisms, and carbohydrate metabolism, responses to biotic stimuli, and defense responses.

## 4. Materials and Methods

### 4.1. Biological Material, Experimental Design, and Stress Application

❖Root dehydration assay

Seeds of *Vigna ungiculata* cv. Pingo de Ouro (considered tolerant to water deficit and drought [63,64] were treated with 0.05% (*w*/*v*) Thiram (tetramethylthiuram disulfide) and germinated during 2 d at 25 ± 1 °C and 65 ± 5% of temperature and relative humidity, respectively. Seedlings were transferred to a hydroponic system [63] (Appendix A) with aerated pH 6.6 balanced nutrient solution [65] in a randomized block experimental design, with three biological replicates (Appendix A). Each biological replicate was composed of two individuals. Plantlets were placed in supports in such a way that the roots of the seedlings were completely immersed in the solution (Appendix A). Plantlets were grown for three weeks (V3 development stage) in a greenhouse under a natural photoperiod of approximately 13/11 h light/dark cycle, temperature of 30 ± 5 °C, and 60 ± 10% relative humidity (RH). After this period, root dehydration treatment was initiated by withdrawing the nutrient solution from treated plants (Appendix A). Roots were collected after 25 min (RD25) and 150 min (RD150) after solution removing (Appendix A). The tissue was immediately frozen in liquid nitrogen and stored at −80 °C until RNA extraction. For each treatment, the respective control plants (Cont.25′ and Cont.150′; Appendix A) were maintained in the nutrient solution and subsequently collected.

Throughout this manuscript, the expression contrast for “RD25 vs. Cont.25” was referred to as T25; similarly, the contrast for “RD150 vs. Cont.150” was referred to as T150.

❖Mechanical injury and virus inoculation assays

The experiments involving mechanical injury and inoculation by CABMV (*Cowpea Aphid-borne Mosaic Virus*) or CPSMV (*Cowpea Severe Mosaic Virus*) were conducted under controlled conditions in a greenhouse at the Agronomic Institute of Pernambuco Instituto Agronômico de Pernambuco (IPA; Recife, Pernambuco, Brazil) (Appendix A). The CABMV assay utilized the resistant genotype IT85F-2687 [66,67], while the CPSMV assay employed the resistant genotype BR-14 Mulato [68].

The experimental procedures for both assays were conducted separately but followed similar protocols. Both accessions were sown and grown for three weeks, reaching the V3 development stage, under natural photoperiods and temperatures ranging from 28 to 32 °C (Appendix A). The leaves of the youngest trifoliate were mechanically injured using carborundum (silicon carbide) to facilitate viral entry into the plant tissue, followed by the application of the viral inoculum (Appendix A).

Two post-injury/inoculation time points were implemented for each assay (Appendix A): 60 min and 16 h. Each treatment had its corresponding absolute control or mock (Appendix A). The tissues were immediately frozen in liquid nitrogen and stored at −80 °C until RNA extraction.

The experimental design was factorial, based on cultivar and post-inoculation period, with three biological replicates (BRs) for each control and treatment (Appendix A). Each BR consisted of five plants. All treatments were carried out in isolated areas to avoid cross-contamination through volatile compounds released by plants.

Differential gene expression in the two virus-related assays was induced by a combination of stress factors: mechanical injury and viral inoculation. Plant viruses cannot initiate infection independently—they require a vector or specific agricultural practices to breach the plant cell wall. According to Barna and Király [69], plant viruses lack specific cellular receptors, unlike bacteriophages and animal viruses. Therefore, the combination of “mechanical injury and viral inoculation” was employed to mimic natural infection processes.

Throughout this study, the expression contrasts for the mechanical injury (MI) and viral inoculation (CABMV or CPSMV) assays were defined as follows:MI + CABMV (60 min) vs. control: MI_CABMV60′;MI + CABMV (16 h) vs. control: MI_CABMV16h;MI + CPSMV (60 min) vs. control: MI_CPSMV60′;MI + CPSMV (16 h) vs. control: MI_CPSMV16h.

### 4.2. RNA Extraction, cDNA Synthesis and Sequencing

Total RNA was isolated using the “SV Total RNA Isolation System” kit (Promega, Madison, WI, USA) following the manufacturer’s protocol, including a DNAse treatment. The concentration, purity, and integrity of total RNA extracted were evaluated by Qubit (Thermo Fisher Scientific, Waltham, MA, USA), Nanodrop (Thermo Fisher Scientific, Waltham, MA, USA), and 1.5% agarose gel (80 V, 120 A for 40 min) stained with Blue-green Loading Dye (LGC Biotechnology, São Paulo, SP, Brazil), respectively. After evaluation by Agilent 2100 Bioanalyzer (Agilent Technologies, Santa Clara, CA, USA), only samples with an RNA integrity number (RIN) ≥ 8.0 were sequenced. The cDNA synthesis was performed using 1 μg of total RNA and Oligo (dT) primers, following the recommendations supplied by GoScript™ Reverse Transcription System Kit (Promega, Madison, WI, USA).

An “RNAm TruSeq^®^ Stranded LT-Set A” kit (RS-122-2101) (Illumina, San Diego, CA, USA) was employed in messenger RNA purification and cDNA library construction according to the manufacturer’s instructions. Paired-end reads 100 bp in length were generated via the Illumina HiSeq 2500 system, using the following kits: “HiSeq^®^ Rapid PE Cluster Kit v2” (PE-402-4002); “SBS Kit v2” (200 Cycle; FC-402-4021); and “TruSeq^®^ Stranded mRNA LT-Set A” (RS-122-2101). All sequencing steps were performed at the Center for Functional Genomics, University of São Paulo (São Paulo, Brazil).

### 4.3. RNA-Seq Libraries Assembly and Differential Expression Analysis

The 12 RNA-Seq libraries sequenced for the root dehydration assay were combined with the 12 libraries from the “mechanical injury and viral inoculation by *Cowpea Aphid-borne Mosaic Virus* (CABMV)” experiment and the 12 libraries from the “mechanical injury and viral inoculation by *Cowpea Severe Mosaic Virus* (CPSMV)” experiment. This combined assembly facilitated the generation of longer and more robust transcripts. A total of 72 replicates (12 biological replicates for root dehydration, 12 for CABMV, and 12 for CPSMV, each with 2 technical replicates) were processed through the pipeline, as briefly outlined below.

Initially, the Trimmomatic v0.36 package was used to remove adapters and low-quality sequences [70]. Reads were trimmed from the 3′ end to ensure a Phred score of at least 30. Illumina sequencing adapters were removed, and only reads with a minimum length of 32 bp were retained. Subsequently, de novo transcriptome assembly was performed using Trinity software 2.0.4 [71], as described by Haas et al. [72]. The quality of the assembled transcriptomes was assessed by evaluating the N50 of the resulting libraries, along with the count and distribution of contig lengths.

Notably, differential expression analysis was conducted independently for each assay (root dehydration; mechanical injury + CABMV inoculation; mechanical injury + CPSMV inoculation). This analysis was performed using the edgeR tool [73] implemented within the Bioconductor package [74]. Transcripts with Log2FC values between −1 and 1, *p* < 0.05, and FDR < 0.05 were considered differentially expressed.

Hierarchical clustering of VuTLPs was carried out with CLUSTER 3.0 [75], and the *heatmap* was visualized using TreeView 2.0 [76]. In addition, Venn diagrams were also generated using the software program Venny 2.0 [77].

### 4.4. Cowpea TLP (VuTLP) In Silico Characterization

The in silico analysis was carried out using sequences of cowpea RNA-Seq libraries obtained under CABMV or CPSMV inoculation or under root dehydration (Appendix A). VuTLPs were predicted based on sequence homology searches, using sequences with thaumatin domains previously characterized [78]. An outline of the annotation strategy is presented in Appendix A. For the identification of VuTLP candidates, the alignments were carried out against CpGC database using tBLASTn [79] (Appendix A) with a cut-off value of 1 × e^−5^. VuTLP candidates were annotated against NCBI and UniProt and analyzed for their score, e-values, sequence size, and presence of conserved thaumatin domains (CTDs), as shown in Appendix A.

All sequences were screened for conserved motifs, with the aid of rps-BLAST on CD-search tool [80] (Appendix A). Only orthologs presenting the expected features (CTDs and motifs) were translated into proteins and considered for subsequent evaluation (Appendix A).

Aiming to enrich the characterization of predicted VuTLPs, the identified amino acid sequences were further characterized including by the following analyses (Appendix A):Determination of the putative isoelectric point and molecular weight using ProtParam [81] (https://web.expasy.org/protparam/, accessed on 25 August 2024); Appendix A;Prediction of signal peptide for each VuTLP with SignalP 4.1 Server [82] (http://www.cbs.dtu.dk/services/SignalP/, accessed on 25 August 2024);Prediction of the subcellular localization with TargetP 1.1 Server [83]; (http://www.cbs.dtu.dk/services/TargetP, accessed on 25 August 2024).

Additionally, we used Clustal Omega [84] (http://www.ebi.ac.uk/Tools/msa/clustalo, accessed on 30 August 2024) (Appendix A) to generate a multiple sequence alignment of full-length VuTLPs compared with previously described TLPs from other plant species, aiming at the localization and comparison of CTDs and residues among TLP sequences. For this step, some homolog proteins denominated “osmotin” and “zeamatin” were also employed (Appendix A), considering that the literature evidences that such proteins present similarities with TLPs [85,86,87].

### 4.5. Distribution of VuTLP Candidates in the V. unguiculata Genome

To determine the genomic distribution of VuTLPs, TLP candidates were aligned against *V. unguiculata* (cowpea) genome V12, available at the Phytozome Database (https://phytozome.jgi.doe.gov/, accessed on 25 September 2024), aiming to map these sequences in virtual chromosomes through the BLASTp tool. This step aimed to infer on the distribution, relative position, and abundance of TLP-coding loci. We considered the best-hit (e-value cut off < 1 × 10−10) to allow for the identification of VuTLPs along the virtual chromosomes. Afterward, the identified mapping positions were used to build a virtual ideogram (Figure 1) regarding the TLP distribution across cowpea pseudochromosomes (n = 11).

### 4.6. Neighbor-Joining (NJ) Analysis and VuTLP Gene Features

For the neighbor-joining (NJ) analysis, 34 VuTLP-coding genes were used. The multiple alignment was manually edited to remove non-aligned ends and uninformative autapomorphies. After that, it was submitted to manual edition and elimination of non-aligned extremities and uninformative autapomorphies. The NJ analysis was carried out using the MEGA 7 [88]. Statistical support of the branches was evaluated by a bootstrap analysis with 2000 replicates.

The gene structure was examined in terms of the exon–intron organization of each VuTLP. Exon–intron visualization was performed using the GSDS 2.0 server [89]. Additionally, other important structural features of the VuTLP genes were determined using the Genestats script, which analyzes 11 distinct structural parameters (https://gist.github.com/darencard/fcb32168c243b92734e85c5f8b59a1c3, accessed on 30 August 2024).

### 4.7. Analysis of VuTLPs Duplication

The Multiple Collinearity Scan toolkit (MCScanX) package [90]—downstream analysis mode: ‘duplicate_gene_classifier’—was applied to classify the origins of the duplicate VuTLP genes of the cowpea genome, into the following:Whole-genome/segmental (i.e., collinear genes in collinear blocks);Tandem (adjacent loci in a genome region);Proximal (gene pairs in nearby chromosomal region but not adjacent);Dispersed (other modes than segmental, tandem, and proximal) duplications.

The strategy utilized by MCScanX to assign the duplication mechanisms can be found in Wang et al. [90].

### 4.8. Ratio of Synonymous (Ks) and Non-Synonymous (Ka) Substitutions per Site for Tandem Duplicated Genes

The ratio of Ka to Ks (Ka/Ks) was used to determine the selection pressure among tandem and segmental duplication events. ClustalW 2.0 software [91] first aligned the full-length coding sequences of tandemly and segmentally duplicated VuTLPs genes. Then, the Ka and Ks rates were calculated with the standard genetic code table by the Nei–Gojobori method (Jukes–Cantor model) implemented on MEGA 7 [88]. A Ka/Ks value less than 1 indicates purifying selection, where deleterious mutations are eliminated, preserving the amino acid sequence. A value of 1 suggests neutral selection, with no selective advantage or disadvantage for mutations. Conversely, a Ka/Ks greater than 1 implies positive selection, favoring mutations that lead to new or altered protein functions.

### 4.9. Candidate Cis-Regulatory Element Analysis

Cowpea promoter regions (up to 1.0 kb upstream of TLP-coding genes) were downloaded from Phytozome database v8.0 (https://phytozome.jgi.doe.gov, accessed on 2 September 2024). The motifs (candidate cis-regulatory elements) in each promoter were revealed through MEME v5.0.3 software [92] (http://meme-suite.org/tools/meme, accessed on 2 September 2024). The software reports an e-value for each identified motif and gives an estimate of the number of expected motifs found by chance if the input sequences were shuffled. In the present simulation, an e-value < 10^−2^ was adopted as the cut-off for characterization of bona fide cis-regulatory element candidates. The maximum number of motifs searched in the present analysis was 10. The motifs analyzed exhibited between 6 and 50 nucleotides in length.

After MEME software analysis, TomTom v4.11.2 software [93] (http://meme-suite.org/tools/tomtom, accessed on 2 September 2024) was used with the JASPAR database (file:JASPAR2018_CORE_plants_non-redundant) to annotate putative cis-regulatory elements. JASPAR searches one or more queries (candidate cis-regulatory element) against annotated motifs ranked by *p*-value (cut-off < 10^−2^). The q-value (false discovery rate; cut-off < 10^−2^) of each match is also reported. In this work, the presented identities of the cis-regulatory element candidates were associated with the target motif that exhibited the most significant *p*-value.

### 4.10. qPCR Analysis

For validation of the RNA-Seq data of the different experiments, RT-qPCR analyses were performed with treatments and control for each time sampled. For each time point, three biological replicates, each with three technical replicates, were analyzed to ensure statistical reliability. The qPCR reactions were performed on the CFX96 Touch Real-Time PCR Detection System (Bio-Rad, Hercules, CA, USA), using SYBR Green detection. The PCR program was adjusted to 95 °C for 2 min, followed by 40 cycles of 95 °C for 15 s and 60 °C for 1 min. After amplification, dissociation curves were produced (65–95 °C at a heating rate of 0.5 °C/s and acquiring fluorescence data every 0.3 °C) to confirm the specificity of the PCR products.

The amplification efficiency (E = 10(−1/slope of the standard curve)) for all primer pairs was determined from a four-point standard curve generated by serial dilutions of cDNA (10-fold each) in technical triplicates. *β-tubulin-2*, *F-box*, and *UBQ10* were used as reference genes for normalization of MI_CABMV and MI_CPSMV assays while *Actin* and *UE21D* were used as reference genes for normalization of radicular dehydration assay [94]. Primers for VuTLP candidates were designed based on the *V. unguiculata* transcriptome using the Primer3 tool (http://bioinfo.ut.ee/primer3-0.4.0/, accessed on 20 August 2024) under the program’s default settings (Appendix A).

Rest software (standard mode) [95] was used for relative expression analysis of target transcripts. Hypothesis testing (*p* < 0.05) was used to determine whether the differences in target transcript expression between the control and treatment conditions were significant.

## 5. Conclusions

The present study describes the structural diversity and gene expression profile of thaumatin-like proteins in cowpea (VuTLPs) under biotic and abiotic stress conditions. Identification of 34 loci encoding VuTLPs along with the detection of both segmental and tandem duplication events underline the complexity of this gene family in cowpea and its evolutionary dynamics. The prevalence of purifying selection across VuTLPs suggests that these proteins have conserved functions important for plant adaptation to stress. The obtained results showed that different stressors provoke distinct expression profiles in VuTLPs and that the most rapid induction generally occurs within minutes following the onset of stress. Delayed expression in response to root dehydration underlines the possibility of functional specialization associated with certain types of unfavorable conditions. This is further supported by the analysis of promoters linking VuTLPs with various transcription factors, including bHLH, Dof-type, and MYB-related factors, indicating multifaceted participation in response mechanisms. Collectively, these analyses provide an overview of VuTLPs and suggest them as potential targets for genetic engineering strategies in cowpea. Towards this, future research studies should henceforth be directed to the elucidation of VuTLP-mediated molecular pathways and interaction with other stress-related proteins for fullest biotechnological applications in crop improvement.

## Figures and Tables

**Figure 1 plants-13-03245-f001:**
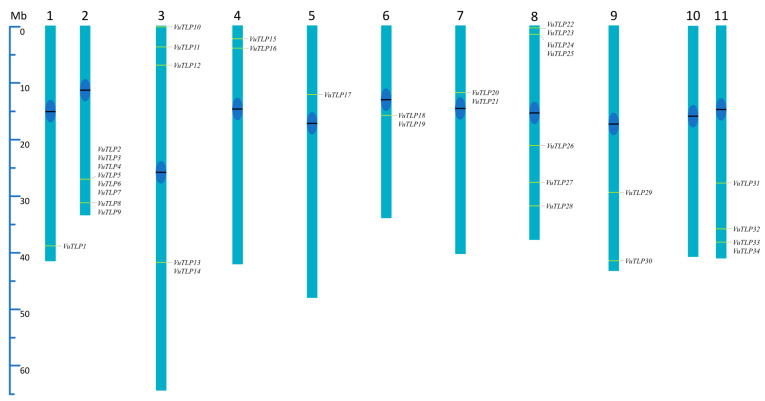
Distribution of cowpea thaumatin-like protein genes in cowpea virtual chromosomes (Chr1 to Chr11). Positions of VuTLP mapped genes or clusters are indicated in yellow. Centromeres are indicated with black bars and dark blue spheres.

**Figure 2 plants-13-03245-f002:**
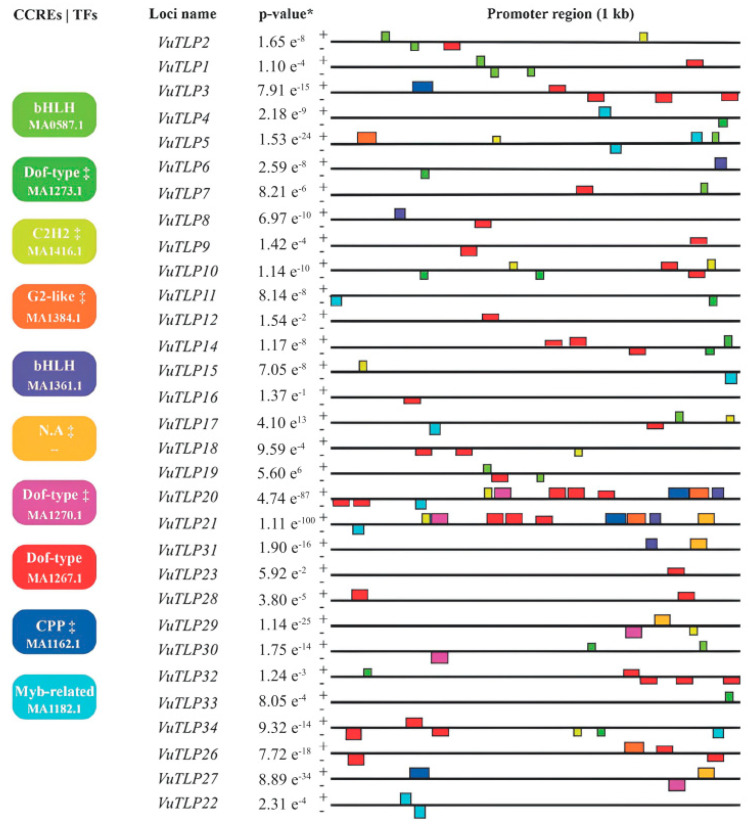
Motif analysis of cis-regulatory elements (CCREs) detected in promoter regions of thaumatin-like protein coding genes. Promoter content and distribution represented by colored rectangles and squares. Colored boxes (CCREs|TFs section) bring information on transcription factors (TFs) associated with the identified motifs, as well as their respective JASPAR ID Matrix. “+” and “-” signals represent sense and antisense strands of the promoter regions analyzed. * MEME combined match *p*-value. ‡ Motifs with statistical significance below of the adopted cut-off (e-value < 10−2). N.A (not annotated motif).

**Table 1 plants-13-03245-t001:** General information on the VuTLP gene duplication mechanisms, protein characteristics, and predicted subcellular localizations.

VuTLP Gene ID	Duplication Mechanism	Signal Peptide	MW (kDa)	pI	REDDD Motif					Subcellular Location
VuTLP1	WGD or segmental	YES	23.27	7.32	R	E	D	D	D	Extracellular
VuTLP2	WGD or segmental	YES	30.75	4.35	R	E	D	D	D	Extracellular
VuTLP3	WGD or segmental	YES	28.48	4.05	R	E	D	D	D	Extracellular
VuTLP4	Proximal	YES	21.57	5.66	R	E	D	D	D	Extracellular
VuTLP5	Tandem	NO	31.03	7.31	R	E	Y	E	D	Plasma membrane
VuTLP6	Tandem	YES	28.11	8.19	Q	E	D	E	D	Extracellular
VuTLP7	Tandem	YES	21.36	6.86	R	E	Y	E	D	Extracellular
VuTLP8	WGD or segmental	YES	21.56	4.04	R	E	D	D	D	Extracellular
VuTLP9	Tandem	YES	24.06	6.74	R	E	D	D	D	Extracellular
VuTLP10	WGD or segmental	YES	23.80	8.76	R	E	D	D	D	Extracellular
VuTLP11	WGD or segmental	YES	21.48	4.29	R	E	D	D	D	Extracellular
VuTLP12	Dispersed	YES	28.55	6.06	R	E	D	N	D	Extracellular
VuTLP13	WGD or segmental	YES	28.72	4.19	R	E	D	D	D	Extracellular
VuTLP14	WGD or segmental	YES	27.88	5.71	R	E	D	D	D	Extracellular
VuTLP15	WGD or segmental	YES	24.89	8.40	R	E	D	D	D	Extracellular
VuTLP16	Dispersed	YES	25.18	7.32	R	E	D	D	D	Extracellular
VuTLP17	WGD or segmental	YES	23.47	7.32	R	E	D	D	D	Extracellular
VuTLP18	Tandem	YES	108.18	5.22	R	E	Y	Q	-	Plasma membrane
VuTLP19	Tandem	NO	13.75	5.46	R	E	D	D	-	Extracellular
VuTLP20	Tandem	NO	27.34	4.77	R	E	D	D	D	Extracellular
VuTLP21	Tandem	NO	27.34	4.77	R	E	D	D	D	Extracellular
VuTLP22	WGD or segmental	YES	30.35	4.21	R	E	D	D	D	Extracellular
VuTLP23	Tandem	YES	30.42	4.24	R	E	D	D	D	Extracellular
VuTLP24	Tandem	NO	29.20	4.65	R	E	D	D	D	Extracellular
VuTLP25	Tandem	YES	22.77	4.30	R	E	D	D	D	Extracellular
VuTLP26	Dispersed	YES	28.05	8.36	R	E	D	D	D	Extracellular
VuTLP27	WGD or segmental	NO	33.67	4.93	R	E	D	D	D	Extracellular
VuTLP28	WGD or segmental	YES	23.48	7.52	R	E	D	D	D	Extracellular
VuTLP29	WGD or segmental	YES	29.89	7.32	R	E	D	D	D	Extracellular
VuTLP30	WGD or segmental	YES	31.53	4.86	R	E	D	D	D	Extracellular
VuTLP31	Dispersed	YES	23.26	4.11	R	E	D	D	D	Extracellular
VuTLP32	Dispersed	YES	24.16	6.80	R	Q	G	D	D	Extracellular
VuTLP33	WGD or segmental	YES	33.65	4.00	R	E	D	D	D	Extracellular
VuTLP34	WGD or segmental	YES	30.44	8.15	R	E	D	D	D	Extracellular

Legend: WGD: whole-genome duplication; MW: molecular weight; pI: isoelectric point.

## Data Availability

The datasets generated for this study can be found in the NCBI BioProject (https://www.ncbi.nlm.nih.gov/bioproject/), with the following accession numbers: Root dehydration|BioSample ID: SAMN14051116; Mechanical injury + CABMV|BioSample ID: SAMN15763372; Mechanical injury + CPSMV|BioSample ID: SAMN15774013.

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
