# Peer review of "Genome-Wide Identification and Stress Responses of Cowpea Thaumatin-like Proteins: A Comprehensive Analysis"

_plants, 2024, doi:10.3390/plants13223245_

Round 1
Reviewer 1 Report
Comments and Suggestions for Authors
The manuscript “Unraveling the Genomic Landscape and Stress Responses of Cowpea Thaumatin-Like Proteins: A Comprehensive Analysis” submitted by Jesús-Pires et al., was carefully reviewed. Thaumatin-like proteins (TLPs) are a highly complex protein family associated with host defense and developmental processes in plants and others. This study started with cowpea and conducted a systematic and thorough comparative analysis of TLP gene families, obtaining some interesting insights that can provide references for further in-depth research on VuTLP members.
The main concerns are as follows:
(1) Landscape genomics generally studies the adaptive genetic variation and evolution of species in response to climate change by analyzing the correlation between genotypes and environmental factors. For your title, what does Genomic Landscape mean and where does it reflect it? It looks very confusing, suggesting changing to a more appropriate title.
(2) Although Figure 1 is the flowchart of this study, it is not suitable for inclusion in the main text. It is recommended to move it into supplementary data.
(3) The MM section, especially 2.6, 2.7, 2.8 etc., are described in too much detail. These analytical methods have references, and it is recommended to refine the content and cite the references.
(4) Figure 3 section, “Promoter analysis associated VuTLPs with bHLH, Dof-type, and MYB-related transcription factors”. The abstract and results sections are described too far fetched and lack further evidence to confirm the content of this sentence.
(5) Which member is the gene ID corresponding to the transcriptome? Should correspond one-to-one with candidate genes. for example “Vu3908|c0_g1_i1, Vu93958|c3_g1_i3, Vu85277|c0_g1_i1, Vu95346|c0_g1_i1”
(6) The Latin name of the virus needs to be italicized.
(7) L169, what is the mean of “100 pb in length”, “bp”?
(8) Some words need to be represented by abbreviations, such as “minutes”, “hours”, “days” etc, search the full text.
(9) Discussion section, lacks descriptions of the validation of endogenous transgenic functions after overexpression and downregulation of candidate genes; Secondly, there is a lack of introduction on which proteins or transcription factors these genes interact with, and the discussion of these contents can provide reference for the subsequent development of related regulatory networks in the system
Reviewer 2 Report
Comments and Suggestions for Authors
Summary
This manuscript delves into the complex world of thaumatin-like proteins (TLPs) in cowpea, exploring how these proteins, part of the PR-5 family, respond to various environmental stresses. The roles of TLPs in cowpea, particularly in the fight against threats such as viral infections and the management of water scarcity, are still being pieced together. In this research, scientists mapped the cowpea genome and pinpointed 34 TLP genes, noting that duplications played a crucial role in the spread and development of these genes. These genes were categorized into three groups according to their structural characteristics. When examining how these genes react under stress, it was found that their activation times and patterns vary. While TLPs generally kicked into gear later during drought conditions, they sprang into action much earlier when the plant faced viral threats. Further examination of the promoter regions of these genes revealed connections to various transcription factors, suggesting that TLPs could play multiple roles in plant defense. The insights gained from this study highlight the pivotal role of TLPs in cowpea resilience and open possibilities for using biotechnology to breed cowpea varieties that are more robust against stress.
General concept comments
1. The manuscript does not adequately justify the significance of studying thaumatin-like proteins (TLPs) or PR5 proteins specifically in cowpea, despite referencing previous research on these proteins in various plant species. A clearer explanation of what makes cowpea TLPs unique or important would strengthen the rationale for this study.
2. Is there any TLP or PR5-like proteins not regulated by abiotic or biotic stress? If so, what is their function/role?
3. The cowpea genotype or cultivar examined in this manuscript appears to differ from those used in the 2011 transcriptomics study by Kido et al. (referenced in Table S1). Clarification on whether these different genotypes or cultivars are comparable would be beneficial for understanding the context and applicability of the findings.
4. Figure 1 and its accompanying description lack clarity regarding the strategies depicted. The shapes and colors used in Figure 1 do not consistently represent their intended meanings. For example, it's unclear whether the green trapezoid symbolizes sequence information. Similarly, does the dark blue represent annotation information, curated sequences, and alignment simultaneously?
5. It would be advantageous to add a table (or revised Table 1) that catalogs the TLP protein numbers with their corresponding gene loci and transcript accession numbers, enhancing the ability for readers to reference specific details effectively. Additionally, for greater clarity in the result sections, it is advisable to reference both the TLP protein numbers and either the gene loci or transcript numbers in the figures and accompanying text. For instance, while Figure 3 currently only displays loci numbers, incorporating the corresponding VuTLP protein numbers would improve its informativeness. Similarly, Figure S4, which presently lists only VuTLP numbers, would benefit from the inclusion of the corresponding gene loci or transcript numbers. Likewise, Figures S6, S7, S9, S10, S11, S12, and S13, which presently feature only transcript numbers, would greatly benefit from the addition of the corresponding VuTLP protein numbers.
Specific comments
1. Line 68-69 (p. 2): How TLPs have biotechnological potential is not well-explained.
2. Clarifying these visual representations will enhance understanding of the figure.
3. Line 218-291 & Figure 1 (p.5): ProtParam is not included in Fig. 1, whereas JVirGel is not explained in the context.
4. The line numbering restarts from 1 on page 8. Please verify if this is consistent with the manuscript's formatting guidelines.
5. Line 109-119 & Supplemental Figure S3: All the transcript numbers are not shown in Figure S3.
6. Is the table spanning pages 12 to 14 intended to be Table 1? It currently lacks a title.
7. Line 316-323: The font size should be revised as it is abnormally large.
Comments on the Quality of English LanguageSentences, mostly in the method section, have grammar issues that need to be revised. For instance,
P.8: “For the Neighbor-Joining (NJ) analysis, used 34 TLP-coding genes.”
p.9: “If Ka/Ks equal ‘1’, means neutral selection (that had no constraint for sequence diver-50 gence)”. And “while Ka/Ks greater than ‘1’ means positive selection (that led to different peptides)”.
Line 81-82 (p.9): A blank follows the sentence “The amplification efficiency… was…”, possibly caused by an accidental keystroke.
Round 2
Reviewer 1 Report
Comments and Suggestions for Authors
The author has provided detailed responses to the questions raised by the reviewers, and the quality of this manuscript has significantly improved. I have no further suggestions.
Author Response
Dear Reviewer #1,
Thank you for your time, effort, and valuable feedback, which provided us with the opportunity to improve our work.
Warm regards,
The authors
Reviewer 2 Report
Comments and Suggestions for Authors
1. In the new Figure S16, replace the lines with arrows to link each shape and clarify the flow direction. The dotted arrow outside the flow or box is confusing and should be removed.
2. The images of Figures 2, S8, S10, and S12 need higher resolution, as some of the text appears blurred.
3. Lines 944-1006: References #1 to #9 and #17 are not left-aligned on each line. Please adjust them for consistent alignment
4. Correct the grammar in Line 720 by revising the sentence from 'For the Neighbor-Joining (NJ) analysis, used 34 TLP-coding genes' to 'For the Neighbor-Joining (NJ) analysis, 34 TLP-coding genes were used."
Author Response
Dear Reviewer #2,
Thank you for your thorough review and attention to our work. All four of your suggestions have been fully addressed. The revised files are available in the supplementary materials, and the modifications in the text are highlighted in blue.
Kind regards,
The authors